# SPARSE MASK REPRESENTATION FOR HUMAN-SCENE INTERACTION

## ABSTRACT

Human-scene interaction is an active research topic with several applications in robotics, virtual experiences, gaming, surveillance, and healthcare. Despite efforts to improve the network architectures to achieve better results or optimize models for faster inference, a crucial aspect of input dimensionality has been somewhat overlooked. This paper introduces Sparse Mask Representation, a simple yet effective approach to enhance the inference speed of human-scene interaction models and improve the model's effectiveness by exploring the sparsity of high-dimensional inputs. Specifically, our method utilizes sparse masks to convert high-dimensional inputs into sparse tensors in a compressed COO format. Our approach not only effectively streamlines computational speed but also eliminates non-useful input information, thereby enhancing overall model performance. We conducted rigorous experiments across three datasets, with a specific emphasis on tasks related to contact prediction and scene synthesis. The results underscore the substantial enhancements realized by our proposed method in terms of accuracy and inference time, surpassing existing state-of-the-art approaches.

## 1 INTRODUCTION

Human-scene interaction explores how humans perceive, navigate, and engage with the environment around them (Hassan et al., 2021). Recently, there has been significant attention on learning the dynamics between humans and the environment (Li et al., 2019; Zhang et al., 2022; Yi et al., 2023). To enhance the modeling and understanding of human pose within diverse environments, researchers have investigated several topics such as human-scene interactions (Hassan et al., 2021; Luo et al., 2023), human-scene synthesis (Zhao et al., 2022b; Shen et al., 2023; Blinn et al., 2021), or human pose contact prediction (Zheng et al., 2022; Huang et al., 2022). Gaining a comprehensive understanding of human posture and interactions with the environment is crucial for various downstream applications (Ye et al., 2022) such as human-robot interaction (Romero et al., 2017; Yi et al., 2022a), realistic virtual experiences (Arsalan Soltani et al., 2017; Zhao et al., 2022b), game animations (Habermann et al., 2021), intuitive interfaces (Zou et al., 2018), advanced surveillance systems (Benfold and Reid, 2009), and healthcare applications (Meng et al., 2023).

In the domain of human-scene interaction, numerous approaches concentrate on generating high-quality scenes based on human contacts and interactions (Hassan et al., 2021; Wang et al., 2022b; Jiang et al., 2022a; Zheng et al., 2022; Yi et al., 2022a; Ye et al., 2022; Wang et al., 2022a; Yi et al., 2023). While the development of complex networks capable of handling the intricacies of scene generation tasks is essential, it also poses challenges in terms of inference speed (Lee et al., 2023) and effectively process the data (Bautista et al., 2022). Yet, many works have acknowledged this problem and thus focused on lightweight architectures, model pruning, or quantization to improve model accuracy and enhance inference speed (Riegler et al., 2017; Tatarchenko et al., 2017; Zhang et al., 2022; Schwarz and Behnke, 2020). However, despite recent developments, current methods still struggle to process complex input structures such as 3D human poses, complex temporal dynamics, or realistic human-scenes interaction.

In this paper, unlike previous methods that primarily focus on designing lightweight models, quantization, model pruning, or diffusion models to enhance human-scene interaction (Hassan et al., 2019; Liu et al., 2022; Jiang et al., 2022b), we propose a solution that focuses on effectively representing the input data. We are motivated by the fact that the input data for human-scene interaction are

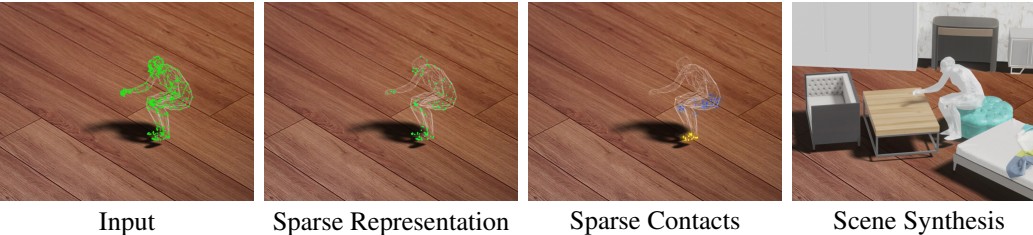

| Input | Sparse Representation | Sparse Contacts | Scene Synthesis |

Figure 1: We present a sparse mask representation to convert high-dimensional inputs into sparse ones for effective contact prediction and scene synthesis.

complex but sparse data structures while having an effective way to represent the input has shown significant improvement in terms of both accuracy and inference speed in other tasks such as affordance learning (Morais et al., 2021) or NeRF-based scene generation (Zhao et al., 2022a; Niemeyer et al., 2022). In particular, we propose Sparse Mask Representation (SMR), a *simple, yet effective method* for human-scene interaction. Unlike other solutions, our simple method utilizes a set of sparse masks to effectively select important information from the input (Figure 1). The sparse marks are then integrated into the human-scene deep backbone by replacing traditional tensor operations with sparse operations. By utilizing sparse operations, our method significantly reduces the computational cost. Intensive experiments show that our method outperforms recent works in contact prediction and scene synthesis tasks while achieving much faster inference speed.

Our key contributions are as follows:

- We introduce sparse mask representation, a simple yet effective method for representing high-dimensional human-scene interaction data.
- We apply our method to different downstream human-scene interaction tasks and demonstrate its effectiveness in terms of accuracy and inference speed.

## 2 RELATED WORK

**Human-scene Interaction.** The human body plays a significant role in facilitating physical interactions (Romero et al., 2017) and in comprehending the contact between humans and their environmental scenes (Li et al., 2019). With advancements in human modeling techniques such as SMPL (Loper et al., 2015), SMPL-X (Pavlakos et al., 2019), MANO (Romero et al., 2017), and FLAME (Li et al., 2017), researchers explore new methods to integrate human skeletons into scenes. For instance, Wang et al. (2017) propose learning affordances from videos to position skeletons in static images. Li et al. (2019) introduce a generative model of 3D poses for predicting plausible human poses within scenes. Several works also focus on collecting or generating data that involve human-scene interactions. Puig et al. (2018) provide a simulated 3D environment where humanoid agents can interact with 3D objects. BEHAVE (Bhatnagar et al., 2022) provides a dataset of real full-body human parameters using the SMPL model while interacting and manipulating objects in 3D with contact points. Based on these datasets, various approaches have been introduced to learn human-scene interaction through scene population (Hassan et al., 2021; Wang et al., 2022b; Jiang et al., 2022a), understand affordances in 3D indoor environments (Li et al., 2019; Kulal et al., 2023; Luo et al., 2023), capture hand and body motions together (Romero et al., 2017; Pavlakos et al., 2019), generate 3D people in scenes (Nie et al., 2022; Wang et al., 2022c), synthesize scene from human motion with diffusion models (Zheng et al., 2022; Yi et al., 2022a; Ye et al., 2022; Wang et al., 2022a; Yi et al., 2023), or track human-object interactions (Blinn et al., 2021; Yi et al., 2022b; Xie et al., 2023). These works contribute to advancing the understanding of human-object interactions, 3D scene generation, and human pose estimation in diverse real-world scenarios (Zhang et al., 2020a; Wang et al., 2022a).

**Lightweight Architecture.** Lightweight methods focus on efficient neural network designs for faster inference and low power consumption in resource-limited scenarios. Network pruning, a prominent approach to achieve this, has been exemplified by (Han et al., 2015; Chakraborty et al., 2018), to eliminate redundancy in large deep networks. Kahatapitiya and Rodrigo (2021) explore

the separation of redundancy and represent it using a smaller parameter set. Quantization techniques (Liu et al., 2018; 2019) leverage lower-bit weight value representations to minimize memory use. Knowledge distillation (Hinton et al., 2015) has emerged as a technique to train lightweight student networks that mimic the behavior of more complex teacher networks. Finally, neural architecture search methods (Guo et al., 2020; Yang et al., 2020; Zoph and Le, 2017; Pham et al., 2018) automatically discover architectures that balance compactness and performance. Lightweight architectures are considered in the context of human-scene interaction to address the complexity of trajectory prediction (Liu et al., 2022; Katariya et al., 2022) or dynamic scene generation (Su et al., 2022; Arad Hudson and Zitnick, 2021). While lightweight networks are appealing, limitations such as reducing modeling capacity and compromising accuracy performance in complex tasks are noteworthy (Cheng et al., 2018). Additionally, they are more prone to overfitting and may struggle to maintain fine-grained information and generalizability (Gupta et al., 2015).

**Sparse Coding.** In addition to lightweight architecture, another direction significantly enhances inference speed is sparse coding. Unlike focusing on architecture design, these methods concentrate on input utilization during learning and inference (Liu et al., 2015). Sparse coding approaches do not modify architectures, instead, they target input format (Choy et al., 2019) and kernel design (Liu et al., 2015; Gray et al., 2017). Specifically, Graham et al. (2018) address inefficiencies in dense convolutional networks by introducing specialized sparse convolutional operations for spatially sparse data and developing submanifold sparse convolution. Chen (2018) directly calculate convolution with a sparse kernel, customize dataflow and memory access instead of converting to matrix multiplication. Graham et al. (2018) develop an implementation of sparse convolution for high-dimensional, sparse input data. Recently, Sylos Labini et al. (2022) present a 1-dimensional blocking algorithm for accelerating sparse matrix multiplication that constructs dense blocks from sparse matrices, providing theoretical guarantees on density.

Although sparse coding works have demonstrated effectiveness in various tasks, they have not been widely applied in human-scene interactions yet, primarily due to limitations in dealing with temporal, contextual dependencies, or the dynamic evolution of interactions over time (Ren et al., 2018). By effectively handling redundant information from inputs, our strategy overcomes these limitations and opens up new possibilities for enhancing real-time interaction prediction and optimizing the efficiency of associated downstream tasks.

## 3 SPARSE MASK REPRESENTATION FOR HUMAN-SCENE INTERACTION

### 3.1 MOTIVATION

Sparse kernels have gained significant popularity in the development of efficient models (Choy et al., 2019; Gray et al., 2017; Graham et al., 2018). However, when compressing models through parameter-space sparsity, the networks still operate on dense tensors, and all intermediate activations within these networks are also dense tensors. This leads to redundancy in the data space during the establishment of computational matrices. Consequently, the full potential of sparse kernels is not maximized. To address this issue, we have a shift in focus towards spatially sparse tensor data, with particular emphasis on sparse high-dimensional 3D inputs and convolution on the surface of 3D objects/humans. In this way, we allow a more efficient utilization of computational resources. By leveraging sparsity in the input, computations between the kernel and input only occur on existing data points, significantly reducing the computational workload based on the input's sparsity. To achieve sparsity in the input, a binary sparse mask is employed to identify which data points would be utilized for the learning process, ensuring the effective utilization of computational resources and enhancing the overall efficiency of the network.

In practice, we observe that increasing the sparsity of the sparse mask results in the loss of input data information and affects the model's performance. Therefore, we utilize multiple sparse masks to generate multiple sparse inputs. As the sparse masks remain unchanged during the learning process, our objective is to assess the contribution of each mask to the task. This assessment allows us to maintain the sparsity of the mask while discarding the masks that do not significantly contribute to the final results, thereby reducing the inference speed and improving model accuracy. Our sparse masks then can be integrated into traditional networks to perform human-scene interaction tasks. Figure 2 shows an overview of our method.

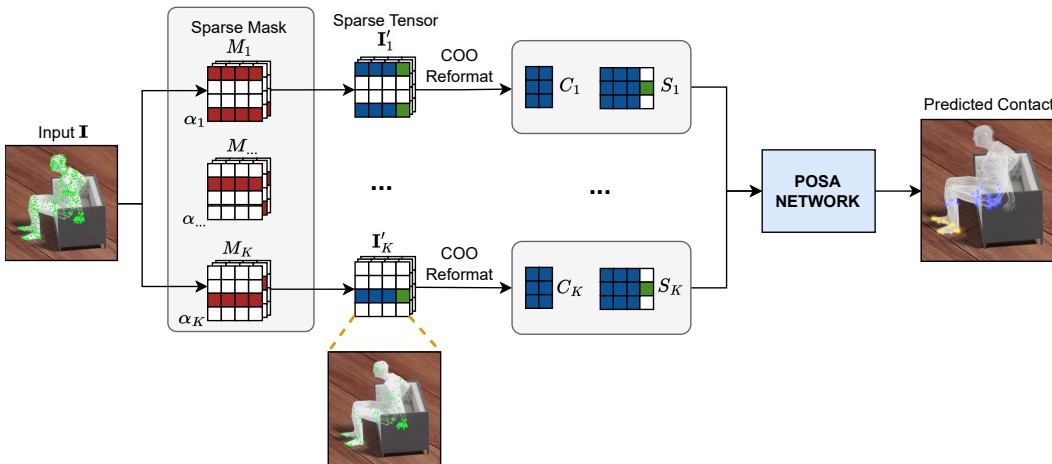

Figure 2: An overview of our method. The red cells denote the non-zero kernel weights and mask values, blue cells denote the coordinate values, green cells denote the non-zero contact values and white cells denote zero values.

## 3.2 SPARSE MASK REPRESENTATION

**Human-Scene Representation.** We follow Hassan et al. (2021) to represent the human-scene interaction. In particular, the human-scene input tensor $I$ is defined as $I = (V, F)$, where $V \in \mathbb{R}^{N_v \times 3}$ is body vertices and $F \in \mathbb{R}^{N_v \times N_c}$ is the contact label of the vertices. $N_v$ is the number of vertices, and $N_c$ is the number of labels.

**Sparse Mask.** Our goal is to convert the human-scene input tensor $I$ into a sparse tensor $I' \in \mathbb{R}^{N_v \times N_S}$ for a more efficient representation ($N_S = N_c + 3$). We define a sparse mask $M \in \mathbb{R}^{N_v \times N_S}$ and calculate $I' = M \circ I$, where $\circ$ denotes element-wise multiplication. Each element in the sparse mask $M$ is sampled from a binomial distribution. The sparsity of $M$ is controlled via a *sparsity ratio* parameter which indicates the non-zero value ratio of the mask. Intuitively, the sparse mask $M$ is a matrix with only 0 or 1 values to mask out the unnecessary information from the input.

In practice, applying only a *single* high-sparsity mask $M$ to the input causes significant information loss hence heavily affecting the effectiveness of the model. To overcome this limitation, we apply $K$ *multiple* sparse masks $\{M_1, M_2, ..., M_K\}$ to the input with the expectation that each sparse mask $M_k$ would learn different important information from the input. We note that each sparse mask $M_k$ is applied independently to the input to obtain the sparse tensor $I'_k$, and $K$ is the hyper-parameter that indicates how many sparse masks we use during training.

**Sparse Mask Representation.** After applying the sparse mask $M_k$ to the input tensor $I$, we obtain a sparse tensor $I'_k = M_k \circ I$ which has a high proportion of zero values. Consequently, the conventional dense representation is inefficient for representing the sparse tensor $I'_k$ during the learning process. Additionally, effectively storing only non-zero values in the sparse tensor facilitates computation (Tew, 2016). To efficiently represent the sparse mask, we find out that the COO format introduced by Chou et al. (2018) is best fitted. since this format is based on the coordinates of non-zero values, and is efficient for neighborhood queries. This representation includes a coordinate matrix $C'_k \in \mathbb{R}^{N'_k \times 2}$ and an associated feature matrix $S'_k \in \mathbb{R}^{N'_k \times N'_S}$ where $N'_k$ denotes the number of non-zero values in $I'_k$. The COO format not only saves memory by removing zero-values from the sparse tensor but also streamlines the computation process for $I'_k$. The sparse tensor $I'_k$ is represented as $I'_k = (C'_k | S'_k)$, where $C'_k$ and $S'_k$ are defined as:

$$C'_k = \begin{bmatrix} b_1 & x_1 \\ \vdots & \vdots \\ b_{N'_k} & x_{N'_k} \end{bmatrix}, S'_k = \begin{bmatrix} s_1^\mathsf{T} \\ \vdots \\ s_{N'_k}^\mathsf{T} \end{bmatrix} \qquad (1)$$

where $(b_i, x_i)$ is the frame index and coordinate of $i$-th feature $s_i \in \mathbb{R}^{N'_S}$.

**Sparse Mask Selection.** Although using a list of sparse masks preserves the model's performance compared to using a single mask, it leads to the fact that some sparse masks capture duplicate information or unnecessary features in the input which may have a negative effect on the results or slow down the inference. To resolve this problem, we define the learnable *mask score* $\boldsymbol{\alpha} \in \mathbb{R}^K$ to *indicate the importance* of each sparse mask. This mask score is calculated based on the contribution of each mask to the final results and the similarity between corresponding masks as follows:

$$\boldsymbol{\alpha}_{(t+1,k)} = \boldsymbol{\alpha}_{(t,k)} + \frac{1}{K-1} \sum_{i \neq k, 1 \leq i \leq K} \left( 1 - \frac{\left\| \boldsymbol{O}_{(t,i)}^{\mathsf{T}} \boldsymbol{O}_{(t,k)} \right\|_{\mathrm{F}}^2}{\left\| \boldsymbol{O}_{(t,k)}^{\mathsf{T}} \boldsymbol{O}_{(t,k)} \right\|_{\mathrm{F}} \left\| \boldsymbol{O}_{(t,i)}^{\mathsf{T}} \boldsymbol{O}_{(t,i)} \right\|_{\mathrm{F}}} \right) \qquad (2)$$

where $\|.\|_{\mathrm{F}}$ is the Frobenius norm; $t$ corresponds to iteration during learning; $\boldsymbol{O}_k$ is the output tensor corresponding to mask $\boldsymbol{M}_k$. Our goal is to compare the differences in distribution between features outputted from different sparse masks to identify which masks mostly produce the same outputs and then discard the redundant ones during the inference process. We note that during training, we utilize $K$ sparse masks and calculate the associated mask scores, while *during testing, we select $\kappa$ masks ($\kappa << K$)* based on the mask score $\boldsymbol{\alpha}$ to use only the useful masks.

**Using Sparse Mask Representation in Deep Layers.** To employ our sparse mask representation in different network layers during training, we simply replace the conventional matrix operations with sparse matrix operations, utilizing input from our sparse mask. This strategy can be applied across different layers, including convolution, batch normalization, pooling, and more, using the COO format (Chou et al., 2018; Choy, 2020) , all without necessitating changes to the network architecture. More details on sparse mask implementation can be found in our Appendix B.

### 3.3 Sparse Network for Human-Scene Interaction

**Contact Prediction.** We train the conditional Variational Autoencoder (cVAE) model, as implemented in POSA (Hassan et al., 2021) for contact prediction. As the input is in the form of a sparse tensor, we replace each layer in Hassan et al. (2021) with a corresponding sparse layer to produce the sparse tensor. This sparse tensor is then passed as the input to the subsequent layer of the network. Note that we only change the original tensor to our sparse tensor, while keeping the whole network unchanged. The Appendix D shows a detailed comparison between our model and POSA.

**Scene Synthesis.** After predicting the contact labels of body vertices in each frame by integrating our sparse tensor into the cVAE model Hassan et al. (2021), we perform the scene synthesis task as a downstream task. We follow the approach outlined by Ye et al. (2022) to conduct the experiment. In particular, we generate objects that make contact with the human body based on the predicted contact points mentioned earlier. The objects that are successfully generated should not penetrate the human body and should align well with the human's intention.

## 4 Experiments

### 4.1 Contact Prediction

**Datasets.** We use the PROXD (Hassan et al., 2019), GIMO (Zheng et al., 2022), and BE-HAVE (Bhatnagar et al., 2022) datasets for contact prediction. In all datasets, the human body is modeled using SMPL-X format (Pavlakos et al., 2019). In the PROXD dataset, the contact labels are obtained from PROX-E dataset (Zhang et al., 2020b).

**Evaluation Metrics**. As in (Ye et al., 2022), the Reconstruction Accuracy and Consistency Score are used for comparing the effectiveness of different methods. We also compare the inference time (second per sample) of all methods on the same NVIDIA Tesla V100 GPU.

**Baselines**. We compare our SMR method with recent works, including POSA (Hassan et al., 2021), ContactFormer (Ye et al., 2022), multi-layer perceptron predictor or bidirectional LSTM (Greff et al., 2016), MIME (Yi et al., 2023), PIAL-Net (Luo et al., 2023), and HOT (Chen et al., 2023). We train our SMR using $K = 10$ masks and keep only $\kappa = 3$ masks with the highest values of mask score $\boldsymbol{\alpha}$ during inference. More implementation details are in Appendix C.

| Methods | Datasets | | | | | | Inference Speed (s/sample) |
| --- | --- | --- | --- | --- | --- | --- | --- |
| | *PROXD* | | *GIMO* | | *BEHAVE* | | |
| | *Reconstruction Accuracy (%)* | *Consistency Score* | *Reconstruction Accuracy (%)* | *Consistency Score* | *Reconstruction Accuracy (%)* | *Consistency Score* | |
| MLP Predictor | 90.84 (+2.85) | 0.892 (+0.089) | 80.7 (+11.4) | 0.801 (+0.142) | 82.5 (+11.3) | 0.724 (+0.149) | 0.11 (↓× 12.2) |
| LSTM Predictor | 90.91 (+2.78) | 0.921 (+0.06) | 83.2 (+8.9) | 0.814 (+0.129) | 80.8 (+13.0) | 0.766 (+0.107) | 0.17 (↓× 18.9) |
| POSA | 91.12 (+2.57) | 0.882 (+0.099) | 89.9 (+2.2) | 0.909 (+0.034) | 89.7 (+4.1) | 0.854 (+0.019) | 0.28 (↓× 31.1) |
| ContactFormer | 91.27 (+2.42) | 0.952 (+0.029) | 90.7 (+1.4) | 0.912 (+0.031) | 91.1 (+2.7) | 0.845 (+0.028) | 0.20 (↓× 22.2) |
| MIME | 90.97 (+2.72) | 0.902 (+0.079) | 89.9 (+2.2) | 0.911 (+0.032) | 90.2 (+3.6) | 0.854 (+0.019) | 0.54 (↓× 60.0) |
| PIAL-Net | 92.04 (+1.65) | 0.953 (+0.028) | 91.1 (+1.0) | 0.934 (+0.009) | 89.9 (+3.9) | 0.864 (+0.009) | 2.97 (↓× 330.0) |
| HOT | 90.9 (+2.79) | 0.966 (+0.015) | 90.3 (+1.8) | 0.900 (+0.043) | 91.7 (+2.1) | 0.821 (+0.052) | 1.12 (↓× 124.4) |
| **SMR (Ours)** | **93.69** | **0.981** | **92.1** | **0.943** | **93.8** | **0.873** | **0.009** |

Table 1: Contact prediction results.

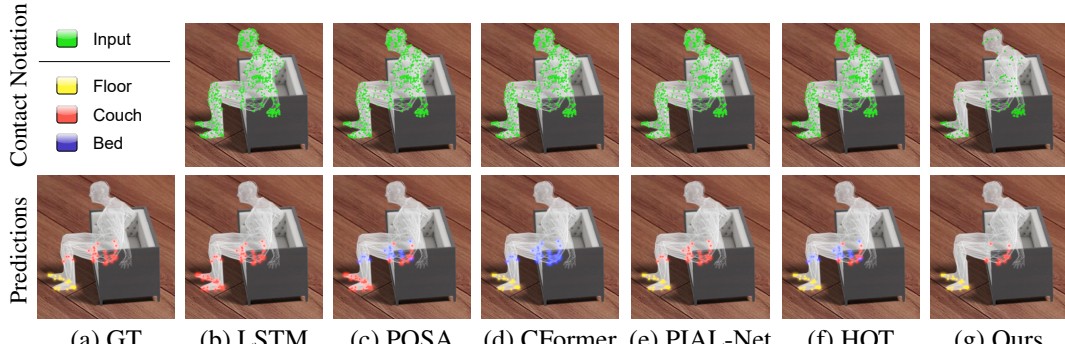

(a) GT  (b) LSTM  (c) POSA  (d) CFormer  (e) PIAL-Net  (f) HOT  (g) Ours

Figure 3: Contact prediction visualization between different methods. We can see that LSTM (b) and POSA (c) show the mismatch between the `Floor` and the `Couch`; ContactFormer (d) and HOT (f) cannot differentiate between `Couch` and `Bed`, while our method shows reasonable predictions.

**Results.** Table 1 shows the comparison between our method and other baselines. This table indicates that our model surpasses all other baselines by a large margin with a reconstruction accuracy of 93.69%, and a consistency score of 0.981 on PROXD dataset. Furthermore, our inference speed is 0.009 second/sample, which *is approximately 12 times faster* than the runner-up.

**Visualization.** Figure 3 shows the qualitative comparison of contact prediction results with different methods. It is notable that our method stands out by achieving accurate contact predictions in both the contact labels and contact location compared to other methods.

### 4.2 SCENE SYNTHESIS

**Datasets.** In the human-scene synthesis task, we use the PROXD (Hassan et al., 2019) and GIMO (Zheng et al., 2022) datasets for conducting experiments as in recent works. Note that BE-HAVE (Bhatnagar et al., 2022) dataset cannot be used in the scene synthesis task since this dataset only has contacts with independent objects, not ones synchronized in a scene.

**Baselines.** We compare our method with recent baselines on the scene synthesis domain, including ContactICP (Besl and McKay, 1992), PosePrior (Moreno-Noguer et al., 2008), SUMMON (Ye et al., 2022), MIME (Yi et al., 2023), and SceneDiffuser (Huang et al., 2023). Our SMR is trained using $K = 10$ masks, then 3 masks with the highest values of mask score $\alpha$ are kept during inference.

**Evaluation Protocol**. We use the non-collision score proposed in Zhang et al. (2020b) as a metric for the scene synthesis task. Furthermore, we perform a user study to compare different methods.

**Results.** Table 2 and Figure 4 provide a comprehensive comparison between scene synthesis results. ContactICP, although exhibiting relatively lower non-collision values, represents an initial approach in this task. Pose Priors (Moreno-Noguer et al., 2008) demonstrates improvements by incorporating pose information, resulting in enhanced reconstruction accuracy. Recent works such as SUMMON (Ye et al., 2022), MIME (Yi et al., 2023), and SceneDiffuser (Huang et al., 2023) show significant advancements, outperforming PosePriors, and achieving notably higher scores on both datasets. However, our method surpasses all other techniques with a recognizable margin, demonstrating a clear improvement in the scene synthesis task.

Table 2: Scene synthesis results. The non-collision score is reported on the PROXD dataset and GIMO dataset.

| Methods | PROXD | GIMO |
|---|---|---|
| ContactICP | 0.654 (+0.282) | 0.820 (+0.131) |
| PosePriors | 0.703 (+0.233) | 0.798 (+0.171) |
| SUMMON | 0.851 (+0.085) | 0.951 (+0.018) |
| MIME | 0.897 (+0.039) | 0.938 (+0.031) |
| SceneDiffuser | 0.914 (+0.022) | 0.942 (+0.027) |
| **SMR (Ours)** | **0.936** | **0.969** |

Table 3: Comparison between different sparse representation methods on PROXD dataset in the contact prediction task.

| Methods | Reconstruction Accuracy (%) | Consistency Score | Inference Speed (s/sample) |
|---|---|---|---|
| POSA | 91.12 (+2.57) | 0.882 (+0.099) | 0.28 (↓× 31.1) |
| ME | 83.61 (+10.08) | 0.797 (+0.184) | **0.008** (↑× 1.13) |
| EsCoin | 69.78 (+23.91) | 0.721 (+0.260) | 0.17 (↓× 1.89) |
| pSConv | 90.24 (+3.45) | 0.825 (+0.156) | 0.084 (↓× 9.33) |
| 1-D Blocking | 88.77 (+4.92) | 0.912 (+0.069) | 0.15 (↓× 16.7) |
| **SMR (Ours)** | **93.69** | **0.981** | 0.009 |

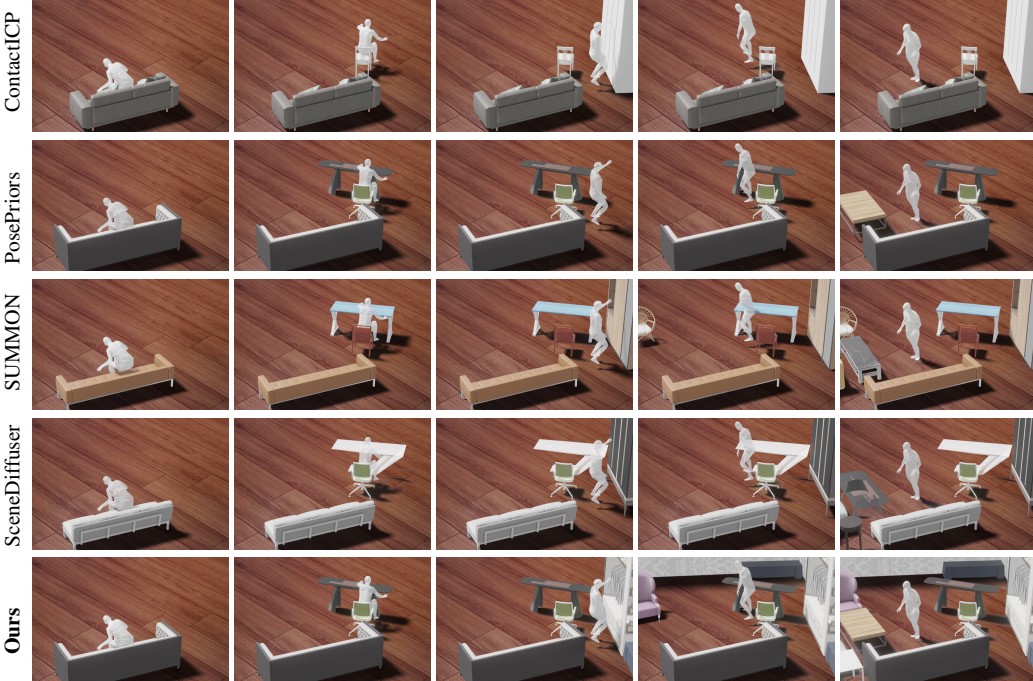

Figure 4: Scene synthesis visualization between different methods. Our method stands out by efficiently utilizing predicted contacts to produce more reasonable and comprehensive scenes.

**User Study.** We conduct a user study with 40 participants from various backgrounds. In this study, participants are presented with a choice between our proposed SMR and current state-of-the-art models, displayed side by side. Both sets of samples are generated using the PROXD test set. This process is repeated five times for each model and the user scores are from 1 to 5. There are two judgment criteria: *(i)* "Naturalness" identifies if the position and orientation of facilities are generated properly in the scene and matched with the human poses or not, and *(ii)* "Non-Collision" shows if the generated object collides with human motions. The results in Figure 5 show that, in most instances, our method is the preferred choice over the compared models. More qualitative results can be found in our Demonstration Video.

### 4.3 COMPARISON WITH OTHER SPARSE REPRESENTATION METHODS

**Baselines.** We compare the effectiveness of the proposed method with four other sparse representation works: ME (Choy et al., 2019), EsCoin (Chen, 2018), pSConv (Kundu et al., 2019), and 1-D Blocking (Jin et al., 2014).

**Implementation.** We use the baseline POSA (Hassan et al., 2021) as the network for contact prediction and report the results in terms of both accuracy (Reconstruction Accuracy and Consistency Score) and inference speed (second/sample).

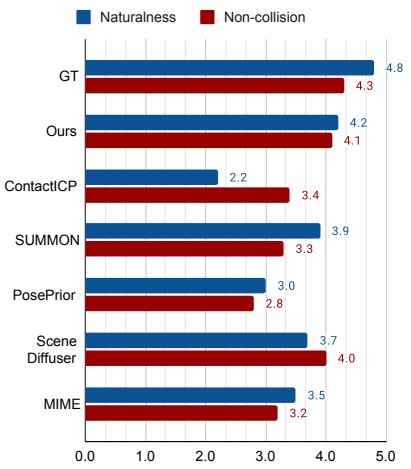

Figure 5: The user evaluation of our method, ground-truth (GT), and other baselines.

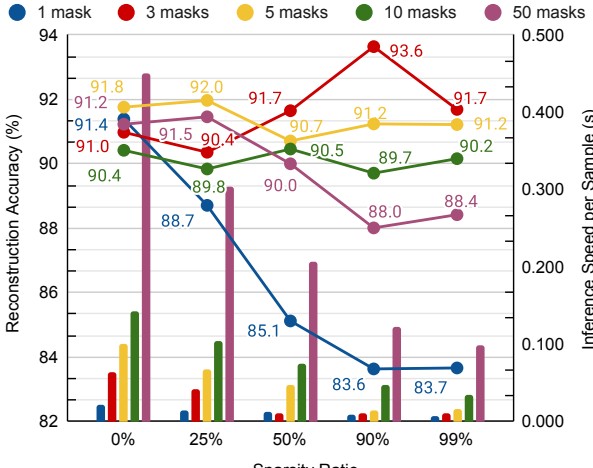

Figure 6: Effectiveness of models with different sparsity ratios and the number of masks.

| Test Cases | Criteria | | | | | |
|---|---|---|---|---|---|---|
| | #Avg. Vertices with contacts↓ | #Avg. Vertices need to predict ↓ | Correct Vertices prediction (%)↑ | Reconstruction Accuracy (%)↑ | Consistency Score↑ | Inference Speed (s/sample)↓ |
| Original Input | 121 | 655 | 90.31 | 91.12 | 0.882 | 0.28 |
| Keep all 50 masks | 107 (↓× 1.13) | 603 (↓× 1.08) | 88.73 (- 1.58) | 89.46 (- 1.66) | 0.935 (+ 0.053) | 0.451 (↑× 1.61) |
| Keep only 01 mask | 12 (↓× 10.08) | 66 (↓× 9.92) | 54.67 (- 35.64) | 83.61 (- 7.51) | 0.763 (- 0.119) | **0.008** (↓× 35.0) |
| Keep only 03 masks | 41 (↓× 2.95) | 66 (↓× 9.92) | **95.65** (+ 5.34) | **93.69** (+ 2.57) | 0.981 (+ 0.099) | 0.009 (↓× 31.1) |
| Keep only 10 masks | 48 (↓× 2.52) | 72 (↓× 9.1) | 92.07 (+ 1.76) | 90.80 (- 0.32) | **0.989** (+ 0.107) | 0.143 (↓× 1.96) |

Table 4: Redundant information analysis by selecting the masks based on mask score $\alpha$. POSA network (Hassan et al., 2021) is used as the backbone.

**Results.** Table 3 presents the performance of different sparse representation methods. We can see that our method achieves the highest accuracy compared to all the other sparse coding baselines. In terms of inference speed, our method is only slower than ME (Choy et al., 2019) (0.009 second/sample vs. 0.008 second/sample) while our accuracy is 10.08% higher than ME.

## 4.4 SPARSE MASK ANALYSIS

**Sparsity ratio and the number of sparse masks.** Figure 6 illustrates the correlation between reconstruction accuracy and inference speed of our method under different values of sparsity ratio and the number of sparse masks $K$. We note that $K = 50$ masks are used during training. During inference, we consequently only select $\kappa$ masks based on the value of mask score $\alpha$. We can see that using $\kappa = 1$ mask leads to faster model performance, however, this also significantly reduces accuracy due to the loss of input information. In contrast, employing multiple sparse masks helps retain essential information and improves the overall model performance. Overall, the experiment in Figure 6 shows that using $\kappa = 3$ masks with 90% sparsity ratio during the inference brings the balance of the accuracy and inference speed.

**How do sparse masks help reduce input information?** Our sparse masks work as a filter to reduce non-useful information in the human-scene input data. In particular, the sparse mask reduces the vertices in human-scene representation and hence, influences inference speed and accuracy. Table 4 illustrates how sparse masks help reduce non-useful input information. The "Original Input" uses all vertices as input; the "Keep only 01 mask" setup uses only $\kappa = 1$ mask during inference. Similarly, we set up our method with 3 masks, 10 masks, and all 50 masks for the inference process, respectively. Note that, our method is trained with $K = 50$ sparse masks, each with a 90% sparsity ratio. The masks are kept based on the mask score $\alpha$ illustrated in Section 3.2. As shown in Table 4, the model using just 1 sparse mask reduces vertex processing requirements by 90%, significantly

enhancing inference speed but causing a $7.51\%$ accuracy drop compared to the Original Input setup. With 50 masks, our SMR approach maintains accuracy but increases inference time since too many masks are used. Using the mask score $\alpha$, we can remove non-useful masks and retain only 10 or even 3 informative masks during inference. We see that using only 3 masks during inference helps reduce the verticle input while increasing the accuracy and reducing the inference speed.

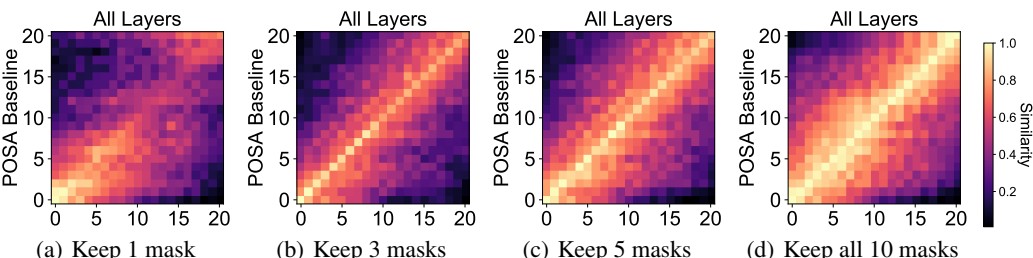

(a) Keep 1 mask     (b) Keep 3 masks     (c) Keep 5 masks     (d) Keep all 10 masks

Figure 7: Similarity between outputs of intermediate layers when different numbers of masks are kept during inference.

**Sparse Masks Selection.** Figure 7 presents the similarity between features of POSA baseline (Hassan et al., 2021) and features of our SMR model when we keep 1, 3, 5, and all 10 sparse masks during the inference. We train the SMR model with $K = 10$ sparse masks, each mask has a sparsity ratio of $90\%$ in this experiment. The mask score $\alpha$ is used to rank and choose useful masks during inference. To compare feature similarity maps, we pass test samples of PROXD dataset (Hassan et al., 2019) to both POSA and our SMR model with the corresponding number of masks. Then, we extract the features from each layer and use the Euclidean distance to compute similarity. While features extracted from the POSA Network remain unchanged in all setups, features of our SMR change when the number of sparse masks is changed. We can see that in Figure 7(a), using only 1 mask with the highest mask score $\alpha$ only maintains feature similarity at abstract layers and the dissimilarity significantly increases in later layers (lightens in early layers and darkens in latter ones). Using 3 masks (Figure 7(b)) or 5 masks (Figure 7(c)) shows good feature similarity within corresponding masks (most features show high similarity in their corresponding layers). This behavior shows that the representations extracted from each layer in our model are distinctive, highlighting how our proposed method handles redundant information compared with all features from the setup that does not use the mask score $\alpha$ to select the useful masks (Figure 7(d)).

## 5 DISCUSSION

We have presented sparse mask representation, a simple yet efficient approach for representing complex human-scene interaction data. Our goal is to expedite the inference process and enhance network performance by reducing redundant information. We have employed our method across various downstream tasks, such as contact prediction and scene synthesis, demonstrating its effectiveness in terms of both accuracy and inference speed.

Although our method shows potential to improve the human-scene interaction task, it does have limitations. First, since our method involves processing the input data using multiple masks, the training time of our model is typically longer than that of the baseline network due to the large number of random masks being used. We note that our strategy currently sacrifices the training time for the inference time. Second, it is challenging to apply our method to recent diffusion works for scene synthesis as the network for diffusion models is relatively simple and the training of diffusion models involves adding noise, which is not compatible with our strategy for effectively representing the input data. Finally, choosing the right sparsity pattern or sparsity ratio can impact the quality of the representation, which requires parameter tuning for a specific task. Failure to make an appropriate selection regarding the number of sparse masks or the sparsity ratio may result in the presence of redundant sparse masks or information loss in inputs, and vice versa. Both cases can lead to inaccurate contact predictions, a primary cause of failure in the synthesis of the existence, position, and orientation of objects in the generated scene. Please refer to Section F in our Appendix for a more comprehensive examination of instances where these failures occur.

There are several avenues for future research from our work. First, developing methods that dynamically adjust the sparsity ratio during inference based on real-time context could improve the flexibility of our approach. Second, extending our method to other related tasks such as action recognition, pose estimation, or object manipulation in dynamic scenes could reveal its potential in a wider range of applications. Finally, applying our method to tiny hardware architectures could yield more meaningful real-world applications.

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

# A   RELATED WORK RECAP

## A.1   CONTACT PREDICTION

Table 5 shows key differences between our work and other recent state-of-the-art methods in the contact prediction task, including POSA (Hassan et al., 2021), ContactFormer (Ye et al., 2022), MIME (Yi et al., 2023), PIAL-Net (Luo et al., 2023) and HOT (Chen et al., 2023). Notably, our proposed SMR achieves competitive results while utilizing significantly fewer parameters compared to recent approaches (approximately 10.7% of MIME, 1.9% of PIAL, and 0.3% of HOT).

| Method | Key Differences | Inputs | # Param |
|---|---|---|---|
| POSA | Use cVAE to sample the contact label, conditioned on the human body vertices. Graph Neural Network is designed for extracting features of each vertex. | Human motions | 1,883,080 |
| Contact Former | Add a transformer layer after the POSA to extract temporal information from a pose sequence, enhancing prediction consistency. | Human motions | 17,324,496 |
| MIME | Leverage POSA to label the contact of vertices. | Human motions | 1.883,080 |
| PIAL | Use Transformer to extract features from images. then establish correlations between the interactable features from diverse images, and finally mine the interactive affinity representation to predict interactive regions. | Images | 10,490,431 |
| HOT | Use CNN to extract features and employ two branches, one for inferring attention masks for body parts and another for extracting contact features. | Images | 50,337,424 |
| **Ours** | Sparse coding the inputs with the compressed COO format to reduce redundant information, with the goal of improving both processing speed and network performance. | Sparse human motions | 200,795 |

Table 5: Key differences between our SMR method with other baselines in contact prediction task.

## A.2   SCENE SYNTHESIS

Table 6 provides a comparison between our method and recent baselines in the scene synthesis task, including MIME (Yi et al., 2023), SceneDiffuser (Huang et al., 2023) and SUMMON (Ye et al., 2022). Despite using a significantly smaller number of contact points and parameters, our SMR method still generates more coherent and inclusive scenes than the other methods. Additionally, our SMR adheres to the algorithm outlined by SUMMON, which does not yield any parameters for the scene synthesis step.

# B   SPARSE LAYERS OPERATION

In this section, we further explore the construction of sparse layers, which enables us to leverage sparse representations for efficient computation. Recall that the sparse layer takes the COO-based sparse input $(\boldsymbol{C}_k, \boldsymbol{S}_k)$ and produces a sparse output $(\boldsymbol{C}'_k, \boldsymbol{S}'_k)$ where $\boldsymbol{C}_k \in \mathbb{R}^{N_k \times 2}, \boldsymbol{S}_k \in \mathbb{R}^{N_k \times N_S}, \boldsymbol{C}'_k \in \mathbb{R}^{N'_k \times 2}, \boldsymbol{S}'_k \in \mathbb{R}^{N'_k \times N'_S}$. Note that $N_k$ and $N'_k$ are the number of non-zero values

| Method | Key Differences | Inputs | # Param |
|---|---|---|---|
| MIME | Combine the contact, foot trace on the floor, and generated objects to generate additional objects by maximizing the log-likelihood of these newly generated objects within the scene. | + Human motions + Contacts + 2D floor + 2D motion on floor | 36,067,282 |
| Scene Diffuser | Augment the time-conditional diffusion model with cross-attention to learn the relation between the input trajectory and scene condition for generating scene. | 3D Scene | 22,625,995 |
| SUMMON | Cluster the vertices of each predicted object class into possible contact instances to generate contact objects, then train an autoregressive transformer model to learn the probability distribution of the non-contact objects. | Human motion + Contacts | 0 |
| **Ours** | Utilize a limited number of predicted contacts and vertices for scene synthesis under ContactFormer backbone. | Sparse human motions + Sparse contacts | 0 |

Table 6: Comparision between our SMR method with other baselines in scene synthesis task.

in the input and output, respectively. Additionally, $N_S$ and $N'_S$ denote the number of feature values in $\boldsymbol{S}_k$ and $\boldsymbol{S}'_k$.

## B.1 CONVOLUTION LAYER

The $k$-th reformatted inputs $\boldsymbol{S}_k$ and $\boldsymbol{C}_k$ are passed through the network and interact with sparse kernels $\boldsymbol{W} \in \mathbb{R}^{m \times m}$ via a mapping function. In the context of the convolutional layer, the kernel weights with an indexing matrix $\mathcal{M}_k^n$ of a $k$-th mask at the $n$-th stride can be calculated as follows:

$$\mathcal{M}_k^n = \begin{bmatrix} \hat{\boldsymbol{W}}[c] \mid \boldsymbol{C}_k[c'] \\ \hat{\boldsymbol{W}}[i] \mid \boldsymbol{C}_k[i'] \end{bmatrix}, \quad \begin{matrix} c = (m^2 - 1)/2, \ c' = i' \ \forall i = c \\ i' = \mathrm{idx}(\boldsymbol{C}_k[\boldsymbol{W}, n, i]), \ \hat{\boldsymbol{W}}[i] \neq 0 \end{matrix} \qquad (3)$$

where $\hat{\boldsymbol{W}}$ is the flattened vector of the kernel $\boldsymbol{W}$ and $\boldsymbol{C}_k[\boldsymbol{W}, n, i]$ is the value when kernel $\boldsymbol{W}$ is applied to the $k$-th sparse input $\boldsymbol{C}_k$ over $n$-th stride corresponding to the $i$ element. $\mathcal{M}_k^n$ is then retrieved in $\boldsymbol{C}_k$ and $\boldsymbol{S}_k$ to compute the sparse output $\boldsymbol{C}'_k$ and $\boldsymbol{S}'_k$ using Equation 4.

$$\begin{aligned} \boldsymbol{C}_k'^n &= [\mathcal{M}_k^n[0][1 :]], \\ \boldsymbol{S}_k'^n &= \sum_{i=1} \mathcal{M}_k^n[i][0] \boldsymbol{S}_k[\mathrm{idx}\,(\mathcal{M}_k^n[i][1])] \end{aligned} \qquad (4)$$

Figure 8 demonstrates our implementation of the convolutional layer applied to sparse tensors.

## B.2 LINEAR LAYER

The linear layer applied for the sparse tensor only changes the number of channels in the feature matrix and does not affect the coordinate matrix. With $\boldsymbol{W}_l \in \mathbb{R}^{N_S \times N'_S}$ and $\boldsymbol{b} \in \mathbb{R}^{N'_S}$ are the weight matrix and bias vector of the linear layer respectively, the output of the linear operator is:

$$\boldsymbol{S}'_k = \boldsymbol{W}_l \boldsymbol{S}_k + \boldsymbol{b}, \ \ \boldsymbol{C}'_k = \boldsymbol{C}_k \qquad (5)$$

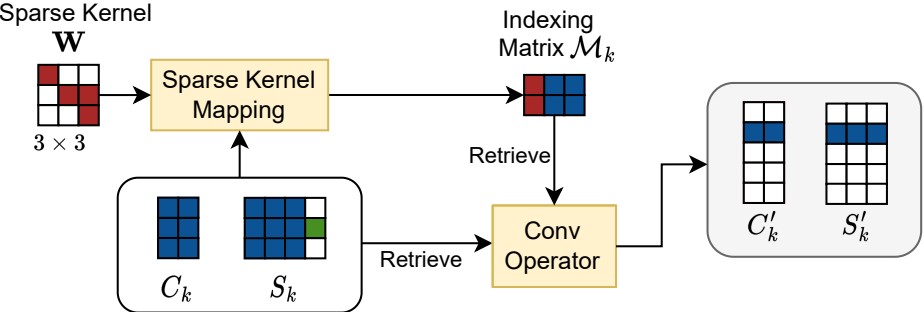

Figure 8: Convolutional layer for sparse tensor. Blue cells denote the non-zero values in the input and output sparse tensor, green cells denote non-zero contact values and red cells denote non-zero kernel weights.

### B.3 POOLING

**Max pooling.** Max pooling layer selects the maximum value in a region for each channel and reduces the shape of the tensor. We define the sparse kernel mapping with the region whose shape is $m \times m$ and stride $n$ for this layer similar to the defined one for the convolution layer. The mapping stores the coordinate of the input in the region and the output each time the region changes location on the sparse tensor:

$$\mathcal{M}_k^n = \begin{bmatrix} \boldsymbol{c} \\ \boldsymbol{c} + \boldsymbol{d} \end{bmatrix}, \quad \forall \boldsymbol{d}[i] \in [0, m), i \in \{0, 1\}, \boldsymbol{c} + \boldsymbol{d} \in \boldsymbol{C}_k \tag{6}$$

where $\boldsymbol{c}$ is the location each time the region pooling slides on the sparse tensor.

$\mathcal{M}_k^n$ is then retrieved in $\boldsymbol{C}_k$ and $\boldsymbol{S}_k$ to compute the sparse output $\boldsymbol{C}_k'$ and $\boldsymbol{S}_k'$ via the following equation:

$$\begin{aligned} \boldsymbol{C}_k^{'n} &= [\mathcal{M}_k^n[0]], \\ \boldsymbol{S}_k^{'n} &= \max_{i=1} \left( \boldsymbol{S}_k[\mathrm{idx}\left( \mathcal{M}_k^n[i][0] \right)] \right) \end{aligned} \tag{7}$$

**Average pooling.** Average pooling is similar to max pooling when we replace the maximum operator with the average one. Figure 9 visualizes how the pooling layer affects sparse tensors.

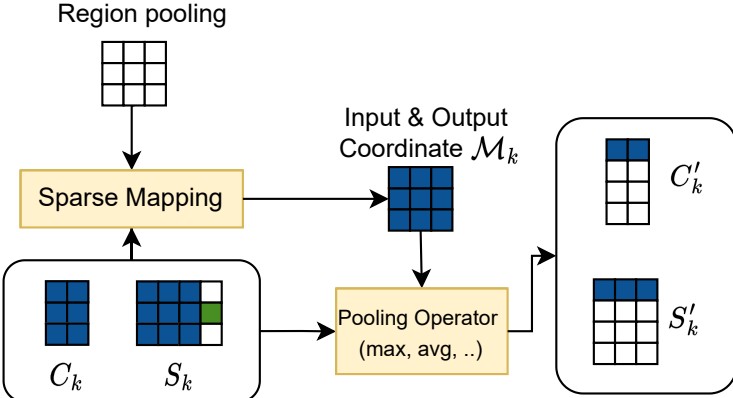

Figure 9: Pooling layer for sparse tensor. Blue cells denote the non-zero values, green cells denote the non-zero contact values, and white cells denote zero values. The matrix $\mathcal{M}_k$ is the same as the one defined in the convolutional layer.

### B.4 NORMALIZATION

**Instance normalization.** Instance normalization normalizes the values along batches and channels. Denote $N_k^b$ as the number of non-zero values in batch $b$, we calculate the mean, standard deviation, and normalized values:

$$\boldsymbol{\mu}_k^b = \frac{1}{N_k^b} \sum_{i:\boldsymbol{C}_k[i][0]=b} \boldsymbol{S}_k[i] \tag{8}$$

$$(\boldsymbol{\sigma}_k^b)^2 = \frac{1}{N_k^b} \sum_{i:\boldsymbol{C}_k[i][0]=b} \left(\boldsymbol{S}_k[i] - \boldsymbol{\mu}_k^b\right)^2 \tag{9}$$

$$\boldsymbol{S}_k^{\prime b} = \frac{\boldsymbol{S}_k^b - \boldsymbol{\mu}_k^b}{\sqrt{(\boldsymbol{\sigma}_k^b)^2 + \epsilon}}, \quad \boldsymbol{C}_k' = \boldsymbol{C}_k \tag{10}$$

**Batch normalization.** Batch normalization normalizes feature values along channels from a whole batch:

$$\boldsymbol{\mu}_k = \frac{1}{N_k} \sum_{i=1}^{N_k} \boldsymbol{S}_k[i] \tag{11}$$

$$\boldsymbol{\sigma}_k^2 = \frac{1}{N_k} \sum_{i=1}^{N_k} \left(\boldsymbol{S}_k[i] - \boldsymbol{\mu}_k\right)^2 \tag{12}$$

$$\boldsymbol{S}_k' = \frac{\boldsymbol{S}_k - \boldsymbol{\mu}_k}{\sqrt{\boldsymbol{\sigma}_k^2 + \epsilon}}, \quad \boldsymbol{C}_k' = \boldsymbol{C}_k \tag{13}$$

**Group normalization.** In group normalization, we divide the features into $G$ group, each group has $N_v/G$ feature values. Then the values in each group are normalized:

$$\boldsymbol{\mu}_{k,g}^b = \frac{1}{N_k^b} \sum_{i:\boldsymbol{C}_k[i][0]=b} \boldsymbol{S}_{k,g}[i] \tag{14}$$

$$(\boldsymbol{\sigma}_{k,g}^b)^2 = \frac{1}{N_k^b} \sum_{i:\boldsymbol{C}_k[i][0]=b} \left(\boldsymbol{S}_{k,g}[i] - \boldsymbol{\mu}_{k,g}^b\right)^2 \tag{15}$$

$$\boldsymbol{S}_{k,g}^{\prime b} = \frac{\boldsymbol{S}_{k,g}^b - \boldsymbol{\mu}_{k,g}^b}{\sqrt{(\boldsymbol{\sigma}_{k,g}^b)^2 + \epsilon}}, \quad \boldsymbol{C}_k' = \boldsymbol{C}_k \tag{16}$$

where $g$ is the group index. When $G = 1$, the group normalization becomes layer normalization instead.

### B.5 NON-LINEARITY LAYERS

The non-linearity layers use activation functions to operate with the sparse tensors such as sigmoid, ReLU, leaky ReLU, ELU, etc. The operation only changes each value in the feature matrix and does not affect the coordinate matrix. With $\boldsymbol{f}_{\text{act}}$ is the activation function, the output is calculated as:

$$\boldsymbol{S}_k' = \boldsymbol{f}_{\text{act}}\left(\boldsymbol{S}_k\right), \quad \boldsymbol{C}_k' = \boldsymbol{C}_k \tag{17}$$

## C    IMPLEMENTATION DETAILS

We build our model by utilizing the POSA backbone by Hassan et al. (2021), and hyperparameters are similar to the setup by Ye et al. (2022). For the ablation study, we set the number of the masks up to $50$ and the sparsity ratio of each mask in set $\{0.1, 0.2, 0.25, 0.3, 0.4, 0.5, 0.6, 0.7, 0.8, 0.9, 0.99\}$. The sparse layers mentioned in Appendix B were implemented using the Minkowski Engine library (Choy et al., 2019) to reduce processing time efficiently. We use NVIDIA Tesla V100 GPU and CUDA 11.6 to train our SMR model for the contact prediction task with a batch size of $64$ for PROXD and GIMO datasets, reduced to $4$ for the BEHAVE dataset. We use Adam optimizer with a learning rate of $0.001$, decayed $10$ times if the metric is not improved within $20$ epochs, and the training process is terminated after $1000$ epochs. The training time depends on the number of masks and the sparsity of each mask, for example, it takes approximately $48$ GPU hours to train our SMR model with $K = 10$ masks and a sparsity ratio of $0.9$ for each mask. The label of each vertex is categorized into $8$ types in the following set: $\{$non-contact, wall, floor, chair, sofa, table, bed, stool$\}$.

## D    NETWORK COMPARISON

| Layer (Type) | Output Shape | Param # |
|---|---|---|
| Linear | [655, 64] | 6,400 |
| GroupNorm | [64, 655] | 128 |
| ReLU | [655, 64] | 0 |
| Linear | [655, 64] | 36,928 |
| GroupNorm | [64, 655] | 128 |
| ReLU | [655, 64] | 0 |
| Linear | [164, 64] | 36,928 |
| GroupNorm | [64, 164] | 128 |
| ReLU | [164, 64] | 0 |
| Linear | [41, 64] | 36,928 |
| GroupNorm | [64, 41] | 128 |
| ReLU | [41, 64] | 0 |
| Linear | [512] | 1,344,000 |
| LayerNorm | [512] | 1,024 |
| ReLU | [512] | 0 |
| Linear | [256] | 131,328 |
| Linear | [256] | 131,328 |
| Linear | [655, 128] | 33,280 |
| GroupNorm | [128, 655] | 256 |
| ReLU | [655, 128] | 0 |
| Linear | [655, 64] | 8,256 |
| GroupNorm | [64, 655] | 128 |
| ReLU | [655, 64] | 0 |
| Linear | [655, 64] | 36,928 |
| GroupNorm | [64, 655] | 128 |
| ReLU | [655, 64] | 0 |
| Linear | [655, 64] | 36,928 |
| GroupNorm | [64, 655] | 128 |
| ReLU | [655, 64] | 0 |
| Linear | [655, 64] | 36,928 |
| GroupNorm | [64, 655] | 128 |
| ReLU | [655, 64] | 0 |
| Linear | [655, 8] | 4,616 |
| **Total # params** | | **1,883,080** |

Table 7: Detailed network architecture of original POSA implemented in Hassan et al. (2021)

| Layer (Type) | Output Shape | Param # |
|---|---|---|
| Sparse Linear | [655, 64] | 3 |
| Sparse Linear | [655, 64] | 640 |
| Sparse GroupNorm | [64, 655] | 16 |
| Sparse ReLU | [655, 64] | 0 |
| Sparse Linear | [655, 64] | 4,616 |
| Sparse GroupNorm | [64, 655] | 16 |
| Sparse ReLU | [655, 64] | 0 |
| Sparse Linear | [164, 64] | 4,616 |
| Sparse GroupNorm | [64, 164] | 16 |
| Sparse ReLU | [164, 64] | 0 |
| Sparse Linear | [41, 64] | 4,616 |
| Sparse GroupNorm | [64, 41] | 16 |
| Sparse ReLU | [41, 64] | 0 |
| Sparse Linear | [512] | 134,400 |
| Sparse LayerNorm | [512] | 128 |
| Sparse ReLU | [512] | 0 |
| Sparse Linear | [256] | 16,416 |
| Sparse Linear | [256] | 16,416 |
| Sparse Linear | [655, 128] | 3,328 |
| Sparse GroupNorm | [128, 655] | 32 |
| Sparse ReLU | [655, 128] | 0 |
| Sparse Linear | [655, 64] | 1,032 |
| Sparse GroupNorm | [64, 655] | 16 |
| Sparse ReLU | [655, 64] | 0 |
| Sparse Linear | [655, 64] | 4,616 |
| Sparse GroupNorm | [64, 655] | 16 |
| Sparse ReLU | [655, 64] | 0 |
| Sparse Linear | [655, 64] | 4,616 |
| Sparse GroupNorm | [64, 655] | 16 |
| Sparse ReLU | [655, 64] | 0 |
| Sparse Linear | [655, 64] | 4,616 |
| Sparse GroupNorm | [64, 655] | 16 |
| Sparse ReLU | [655, 64] | 0 |
| Sparse Linear | [655, 8] | 576 |
| **Total # params** | | **200.795** |

Table 8: Detailed network architecture of our SMR method that use 3 masks.

Table 7 and Table 8 show the comparison between layers (including the function, the output shape when passing to that layer, and its number of parameters) implemented in the architecture of the POSA model and our SMR model. The reported SMR model uses 3 masks during inference, and the sparsity of each is 90%. The main differences between these layers in these models are:

    i. We add the learnable mask score $\alpha$ on the top of SMR architecture to control the contribution of the sparse masks.

    ii. We replace the original dense layers implemented in the POSA model with the same functional corresponding sparse layers mentioned in Appendix B in our SMR model.

The sparse layers with the sparse kernel have a number of parameters less than 8 to 10 times compared with dense layers. This results in about a $\sim 9.4$ times reduction in the total number of parameters in the SMR. In addition, the input for SMR is the sparse body mesh with a $90\%$ fewer vertices, while POSA takes the whole body mesh as input. Therefore, the inference process of our method is significantly faster than the POSA baseline.

## E    EXTENDED ANALYSIS

### E.1    CORRELATION BETWEEN SPARSITY AND NUMBER OF SPARSE MASKS

In Figure 6, we have analyzed the effectiveness of models with varying sparsity ratios and the number of masks using our proposed SMR. For further clarification, we conduct an extended analysis to examine the pattern of accuracy and speed concerning these mentioned parameters. Figure 10 illustrates the results of SMR when we change the number of sparse masks and the sparsity ratio. In particular, Figure 10-a shows the Reconstruction Accuracy, and Figure 10-b demonstrates the corresponding GPU inference time. The results indicate that as the number of masks and the sparsity ratio increase, the inference speed decreases. Additionally, more redundant masks can be established. However, with an appropriate trade-off, state-of-the-art results with efficient inference time can be achieved. We can observe that, with a $90\%$ sparsity ratio and 3 masks in SMR, we achieve a state-of-the-art $93.6\%$ accuracy while still maintaining efficient processing during inference (0.01 second per sample).

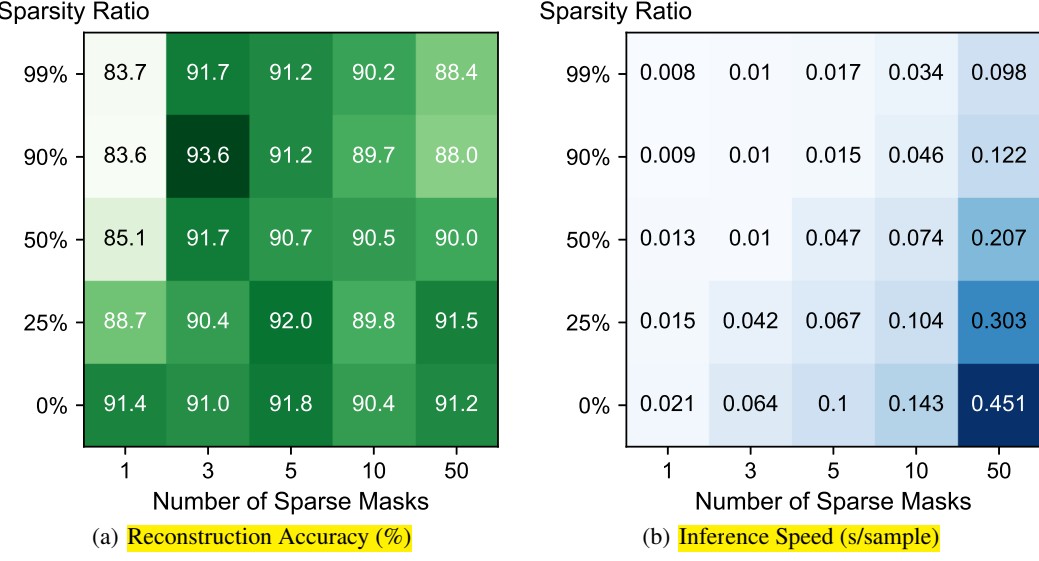

(a) Reconstruction Accuracy (%)             (b) Inference Speed (s/sample)

Figure 10: Reconstruction Accuracy (%) and Inference Speed (s/sample) between different setups.

### E.2    SPARSE MASK REPRESENTATION COMPONENT ANALYSIS

Table 9 below shows some results regarding the contribution of sparse masks and a sparse network. POSA serves as our baseline, and if setups involve masks, three masks are used. It is evident that when we use sparse masks without the COO format, the number of data points remains unchanged, leading to no improvement in speed and, in fact, a decrease in accuracy due to missing information.

Applying a sparse network to original inputs improves speed, but the trade-off for accuracy is noticeable, as discussed in many previous papers. When we apply the COO format to the original inputs, the differences in speed and accuracy are not significant compared to the original baseline. If sparse masks and the COO format are both used without the sparse network, we must retain all zero points, even in the COO format, to match the input shape for the network. Consequently, no improvement in speed is observed, and the model's effectiveness is limited. With our introduced refinement process that works on COO format, we can preserve the performance of the model but the speed improvement is not guaranteed. Ultimately, when we use COO inputs obtained from sparse masks and a sparse network together, with the stored indexes in COO format, the input shape problem can be addressed, achieving optimization in both speed and accuracy.

| Network | COO Format | Sparse Masks | Sparse Mask Refinement | Reconstruction Accuracy (%) | Inference Speed (s//sample) |
|---|---|---|---|---|---|
| Dense | | | | 91.12 | 0.28 |
| Dense | | ✓ | | 85.72 (- 5.4) | 0.28 (↓× 1.0) |
| Sparse | | | | 83.41 (- 7.71) | 0.09 (↓× 3.11) |
| Dense | ✓ | | | 91.02 (- 0.1) | 0.27 (↓× 1.04) |
| Dense | ✓ | ✓ | | 85.46 (- 5.66) | 0.28 (↓× 1.0) |
| Dense | ✓ | ✓ | ✓ | 93.27 (+ 2.15) | 0.28 (↓× 1.0) |
| Sparse | ✓ | ✓ | ✓ | **93.69** (+ 2.57) | **0.01** (↓× 28) |

Table 9: SMR Component Analysis. Results are benchmarked on the PROXD dataset. POSA (Hassan et al., 2021) is the backbone. The kept 3-mask setup is used when Sparse Masks are available.

### E.3 COMPARISON WITH MESH SUBSAMPLING METHODS

Table 10 illustrates the comparison between our proposed SMR and other Mesh Subsampling methods, including typical algorithm-based methods and deep-based ones. Across all cases, the proposed method significantly outperforms other approaches in terms of both speed and accuracy. This result is easily explainable, as most mesh subsampling methods do not adhere to any specific algorithms during the subsampling process to preserve model performance. Additionally, their primary objectives revolve around finding a better discrete representation of a mesh with triangles of equal edge length, rather than catering to human-scene interaction tasks. Besides, some deep-based mesh simplifier methods also take time to finish their sampling process.

### E.4 REDUNDANT INFORMATION ANALYSIS

Figure 11 illustrates the ground truth, results from the Baseline POSA (which takes dense tensors as inputs), and our SMR approach that considers sparse masks. It's evident that vertices in the legs are redundant and are not taken into account in our proposed SMR method. However, POSA still considers this information, leading to incorrect predictions in contact points. The results and visualization imply that our proposed SMR successfully reduces redundant information in the inputs hence increasing the model's effectiveness.

## F FAILURE CASES

Figure 12 shows the failure cases of generated scenes. These results mostly involve unnecessary objects or objects generated with incorrect positions/orientations, making the scenes implausible. This can occur due to the absence of the necessary information or an excess of redundant information needed to generate the scenes when we fix the number of the selected mask $\kappa$. While fine-tuning the mask score $\alpha$ in the inference process to select the top $\kappa$ masks and address the mentioned problem

| Method | Type | Reconstruction Accuracy (%) | Inference Speed (s/sample) |
|---|---|---|---|
| Baseline (POSA) (Hassan et al., 2021) | Algorithm-based | 91.12 | 0.28 |
| Isotropic Remeshing (Alliez et al., 2003) | Algorithm-based | 82.18 (-8.94) | 0.18 (↓× 1.56) |
| Vertex Clustering (Rossignac and Borrel, 1993) | Algorithm-based | 78.67 (-12.45) | 0.13 (↓× 2.15) |
| Incremental Decimation (Ghazanfarpour et al., 2020) | Algorithm-based | 79.45 (-11.67) | 0.09 (↓× 3.11) |
| Neural Mesh Simplification (Potamias et al., 2022) | Deep-based | 85.56 (-5.56) | 0.47 (↑× 1.68) |
| CoMA (Ranjan et al., 2018) | Deep-based | 89.22 (-1.90) | 0.23 (↓× 1.22) |
| **Ours** | - | **93.69** (+ 2.57) | **0.01** (↓× 28) |

Table 10: Result comparison between SMR and other mesh subsampling methods. Results are benchmarked on the PROXD dataset. The compression ratio at each algorithm-based method is maximized until it reaches the maximum edge merging limit.

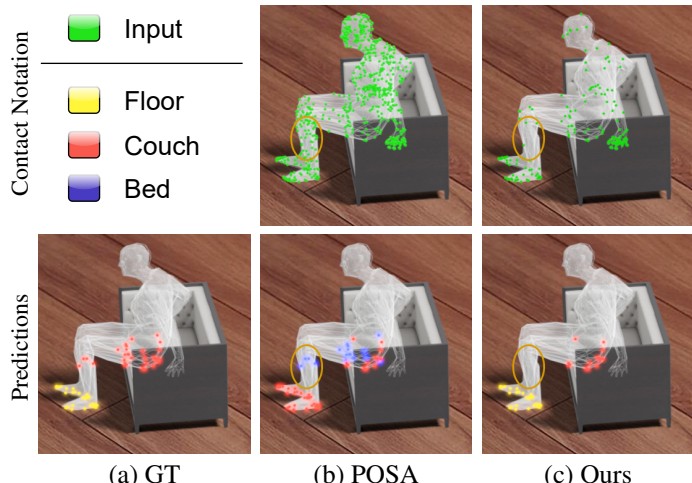

(a) GT      (b) POSA      (c) Ours

Figure 11: Scenario when the human is sitting. The points in the dark yellow circle are the redundant ones when predicting contact in a sitting pose.

does not consume much time, further improvements can be made by learning the optimal value of $\kappa$ directly during training.

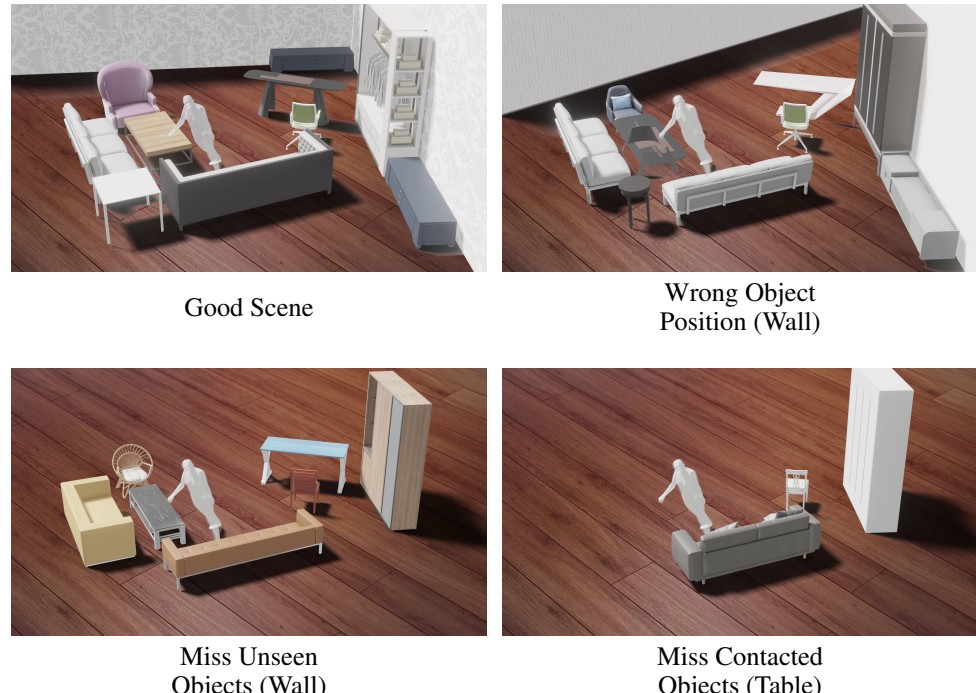

Good Scene

Wrong Object
Position (Wall)

Miss Unseen
Objects (Wall)

Miss Contacted
Objects (Table)

Figure 12: Visualization of a good case scene synthesis compared with failure cases.

