# OpenReview forum: "Sparse Mask Representation for Human-Scene Interaction"
_ICLR.cc/2024/Conference — Submitted to ICLR 2024_

### Official Review · Reviewer_BCwn · 2023-10-18

**Soundness:** 2 fair
**Presentation:** 2 fair
**Contribution:** 1 poor
**Rating:** 3
**Confidence:** 4

**Summary:**

The paper proposes the use of sparse tensors to represent human-scene interaction data. Given a dense tensor input the authors proposed to learn multiple sparse masks. Sparse tensors are created by multiplying the learned masks and the dense input tensor. The sparse masks have a pre-defined fixed sparsity. The authors reused existing dense architecture but converted its dense operations into sparse ones.
The paper shows two applications: contact prediction and scene generation

**Strengths:**

- The input dimensionality in human-scene interaction is rightfully large. Using sparse input for this task is novel and technically sound.
- The paper is easy to read.
- Human-scene interaction is an interesting and important problem.

**Weaknesses:**

- The novelty is limited to sparsifying the input for the human-scene interaction. Using sparse inputs by itself is not new, but it has not been studied for this task before.
- In Figure 6 and Table 4. It seems the model gets worse when using 50 or 10 masks which is strange.Why would using 3 masks be better than 10 or 50 masks? Why would it be even better than using the full dense tensor?
- The paper attributes the COO representation to "Choy 2020". The COO format is much older than that.
- Figure 4 is hard to see. Human bodies are too small.

**Questions:**

- The method is basically learning mesh subsampling. I wonder about how the method compares to classic subsampling methods.
- Did the masking learn any interesting patterns? like which vertices are more relevant for which pose?
- I understand that the method can be faster than POSA, but why would it be more accurate?

---

> ### Author Response · Authors · 2023-11-19
> **Response for Reviewer BCwn (Part 1/2)**
>
> **Q1: The novelty is limited to sparsifying the input for the human-scene interaction. Using sparse inputs by itself is not new, but it has not been studied for this task before.**
>
> > The use of sparse masks to enhance model speed has been investigated and established as a path within Sparse Coding solutions, as outlined in our Related Work. However, as you mentioned, its effectiveness on the human-scene interaction task has not been studied before. Unlike other methods that mostly apply the sparsity for improving inference speed, our method can increase both accuracy and speed. Despite the simplicity, our method outperforms recent work by a significant margin, for example, we outperform MIME [1] with 2.72% higher accuracy and 60 times faster.
>
> **Q2: In Figure 6 and Table 4. It seems the model gets worse when using 50 or 10 masks which is strange. Why would using 3 masks be better than 10 or 50 masks? Why would it be even better than using the full dense tensor?**
>
> >We value the feedback provided by the reviewer. The POSA backbone operates on dense tensor inputs. However, ***not all vertices contribute meaningfully*** to predicting human-scene contacts, leading to redundant information that can induce overfitting during the learning process. Sparse masks aim to mitigate this issue by increasing sparsity and selecting useful information. Nonetheless, an excessively large set of sparse masks can introduce a similar problem of redundant information, which adversely affects results when employing 50 or 10 masks.
>
> >Among these, the kept 3-mask setup refined by our introduced Sparse Mask Refinement technique yields the best results. This refinement process helps eliminate redundant sparse masks, enabling even better performance than the baseline which still suffers from redundant information issues. For further detailed insights, please refer to Section 4.4 'How do sparse masks help reduce input information?'
>
> **Q3: The paper attributes the COO representation to "Choy 2020". The COO format is much older than that.**
>
> >We appreciate the reviewer for pointing this out and concur that the original COO format has been appropriately cited. To the best of our knowledge, the initial representation of a sparse tensor using the COO format is documented by [2] (Chou et al., 2018), and subsequent extensions to operations on sparse tensors are detailed in [3] (Choy et al., 2020). Please do let us know if we missed any citations.
>
> **Q4: Figure 4 is hard to see. Human bodies are too small.**
>
> >We appreciate your feedback. We have enhanced the size of Figure 4.
>
> **Q5: The method is basically learning mesh subsampling. I wonder about how the method compares to classic subsampling methods.**
>
> >We appreciate the reviewer for mentioning the comparison with mesh subsampling methods. We agree that these methods also enhance the inference speed, with much of the mesh subsampling process occurring at runtime.
>
> >The table below illustrates the comparison between our proposed SMR and other Mesh Subsampling methods, including typical algorithm-based methods and deep-based ones. Across all cases, our method significantly outperforms other approaches in terms of both speed and accuracy. This result is easily explainable, as most mesh subsampling methods do not adhere to any specific algorithms during the subsampling process to preserve model performance. Additionally, their primary objectives revolve around finding a better discrete representation of a mesh with triangles of equal edge length, rather than catering to human-scene interaction tasks. Besides, some deep-based mesh simplifier methods also take time to finish their sampling process.
>
> >|Methods|Mesh Subsampling|Reconstruction Accuracy(%)|Speed (s/sample)|
> |:-|:-:|:-:|:-:|
> |Baseline (POSA)|Algorithm-based|91.12|0.28|
> |Isotropic Remeshing [4]|Algorithm-based |82.18(-8.94)|0.18($\downarrow$ 1.56)|
> |Vertex Clustering [5]|Algorithm-based | 78.67 (-12.45)|0.13($\downarrow$ 2.15)|
> |Incremental Decimation [6]|Algorithm-based | 79.45 (-11.67) | 0.09 ($\downarrow$ 3.11)|
> |Neural Mesh Simplification [7]|Deep-based|85.56(-5.56)|0.47($\uparrow$ 1.68)|
> |CoMA [8]|Deep-based|89.22(-1.90) |0.23($\downarrow$ 1.22)|
> |**Ours**|-|**93.69**(+2.57)|**0.01**($\downarrow$ 28)|
> >
> >*Result comparison between SMR and other mesh subsampling methods. Results are benchmarked on the PROXD dataset.*

---

> > ### Author Response · Authors · 2023-11-19
> > **Response for Reviewer BCwn (Part 2/2)**
> >
> > **Q6: Did the masking learn any interesting patterns? like which vertices are more relevant for which pose?**
> >
> > >As indicated in Table 4 of the main submission, the human mesh in the sitting pose comprises a total of 655 vertices, including approximately 121 contact points. Through the utilization of sparse masks and the elimination of unnecessary masks using the Sparse Mask Refinement technique, the total number of vertices is reduced to around 66, with 41 vertices in contact. These contact points primarily originate from the vertices near the buttocks in the sitting pose, providing significant information for predicting contact labels. Conversely, points in other areas of the human body, such as hands, fingers, head, or legs, are irrelevant for the sitting pose and hence redundant, allowing for their removal.
> >
> > >Figure 3 in our main paper demonstrates this case. We have further included Figure 11 in section E.4 of our appendix for a detailed explanation. From these figures, it is evident that vertices in many human joints are redundant and are not taken into account in our proposed SMR method. However, POSA still considers this information, leading to incorrect predictions in contact points. The results and visualization show that our proposed SMR successfully reduces redundant information in the inputs hence increasing the model's effectiveness.
> >
> > **Q7: I understand that the method can be faster than POSA (full tensor), but why would it be more accurate?**
> >
> > >The baseline POSA, which involves a comprehensive tensor setup, includes numerous vertices. Many of these vertices are redundant for the human-scene interaction task and could potentially lead to overfitting. However, by utilizing sparse masks and implementing the Sparse Mask Refinement technique, we effectively mitigate this issue, resulting in superior outcomes compared to the baseline POSA. Figure 3 and Table 4 in our main paper demonstrate this scenario.
> >
> > ## References:
> > >
> > >[1] H. Yi, C.-H. P. Huang, S. Tripathi, L. Hering, J. Thies, and M. J. Black. Mime: Human-aware 3d scene generation. In CVPR, 2023.
> > >
> > >[2] S. Chou, F. Kjolstad, and S. Amarasinghe. Format abstraction for sparse tensor algebra compilers. Proceedings of the ACM on Programming Languages, 2018.
> > >
> > >[3] C. B. Choy. High-Dimensional Convolutional Neural Networks for 3D Perception. Stanford University, 2020.
> > >
> > >[4] P. Alliez, E. C. De Verdire, O. Devillers, and M. Isenburg. Isotropic surface remeshing. In 2003 Shape Modeling International., IEEE, 2003.
> > >
> > >[5] J. Rossignac and P. Borrel. Multi-resolution 3d approximations for rendering complex scenes. In Modeling in computer graphics: methods and applications, Springer, 1993.
> > >
> > >[6] A. Ghazanfarpour, N. Mellado, C. E. Himeur, L. Barthe, and J.-P. Jessel. Proximity-aware multiple meshes decimation using quadric error metric. Graphical Models, 2020.
> > >
> > >[7] R. A. Potamias, S. Ploumpis, and S. Zafeiriou. Neural mesh simplification. In Proceedings of the CVPR, 2022.
> > >
> > >[8] A. Ranjan, T. Bolkart, S. Sanyal, and M. J. Black. Generating 3d faces using convolutional mesh autoencoders. In ECCV, 2018.

---

> ### Author Response · Authors · 2023-11-21
> **A friendly reminder**
>
> Dear Reviewer **BCwn**,
>
> We appreciate your time and feedback on our paper. We have responded to all the concerns you raised. Please let us know if you have any further questions.
>
> Best regards,
> Authors.

---

### Official Review · Reviewer_gK1o · 2023-10-29

**Soundness:** 2 fair
**Presentation:** 3 good
**Contribution:** 2 fair
**Rating:** 5
**Confidence:** 3

**Summary:**

Instead of focusing on optimizing the model architecture, the authors proposed a novel way to enhance the human-scene interaction research from the view of representation.
It is revealed that the input for human-scene interaction is usually of high dimension, which limits the inference speed and effectiveness of the models.
Sparse Mask Representation is thus proposed, exploring the sparsity of the inputs.
Rigorous experiments are conducted on tasks related to contact prediction and scene synthesis.
Results show the effectiveness of the proposed sparse encoding.

**Strengths:**

The authors show that introducing sparse encoding is an effective technique for the improvement of Human-Scene Interaction tasks.
Impressive inference acceleration and model compression are achieved with the proposed method.
Competitive results are shown compared to previous efforts.

**Weaknesses:**

The current version appears to be an application of the Choy, 2020 citation. Clarification on the contribution beyond this should be provided.

As mentioned, the acceleration could be attributed to two factors. First, a sparse body mesh with 90% fewer vertices is used. Second, the sparse network works. Ablation should be conducted on the sparse mesh only and the sparse network only.

**Questions:**

Please refer to the Weaknesses section.

---

> ### Author Response · Authors · 2023-11-19
> **Response for Reviewer gK1o**
>
> **Q1: The current version appears to be an application of the Choy, 2020 citation. Clarification on the contribution beyond this should be provided.**
>
> >- Our method is an input representation technique that centers on a learned set of sparse masks to generate zero-filtered data points.
> >- COO is a matrix reformatting technique used to restructure the inputs. The indexing characteristic of the COO enables us to safely eliminate the mentioned zero points while guaranteeing the incorporation of the remaining data into the network.
> >- Simply speaking, our method learns and ranks the sparse masks, while the COO ensures sparsity implementation. Our experiment shows that we need both the sparse masks and the COO to obtain the best results.
>
>  **Q2: As mentioned, the acceleration could be attributed to two factors. First, a sparse body mesh with 90% fewer vertices is used. Second, the sparse network works. Ablation should be conducted on the sparse mesh only and the sparse network only.**
>
> >We appreciate the reviewer's question. The table below shows some results regarding the contribution of sparse masks and a sparse network. POSA serves as our baseline, and if setups involve masks, three masks are used. It is evident that when we use sparse masks without the COO format, the learning is not effective since there are too many zero points from the input without being removed, leading to no improvement in speed and, in fact, a decrease in accuracy. Applying a sparse network to original inputs improves speed, but the trade-off for accuracy is noticeable, as discussed in many previous papers. When we apply the COO format to the original inputs, the differences in speed and accuracy are not significant compared to the original baseline. If sparse masks and the COO format are both used without the sparse network, we must retain all zero points, even in the COO format, to match the input shape for the network. Consequently, no improvement in speed is observed, and the model's effectiveness is limited. With our introduced Sparse Mask Refinement that works on COO format, we can preserve the performance of the model but the speed improvement is not guaranteed. Ultimately, when we use COO inputs obtained from sparse masks and a sparse network together, with the stored indexes in COO format, the input shape problem can be addressed, achieving optimization in both speed and accuracy.
>
>
> >|Network|COO Format|Sparse Masks|Sparse Mask Refinement|Reconstruction Accuracy (%)|Inference Speed (s/sample)|
> |:-:|:-:|:-:|:-:|:-:|:-:|
> |Dense||||91.12|0.28|
> |Dense||x||85.72(-5.4)|0.28($\downarrow$ 1.0)|
> |Sparse||||83.41(-7.71)|0.09($\downarrow$ 3.11)|
> |Dense|x|||91.02(-0.1)|0.27($\downarrow$ 1.04)|
> |Dense|x|x||85.46(-5.66)|0.28($\downarrow$ 1.0)|
> |Dense|x|x|x|93.27(+2.15)| 0.28($\downarrow$ 1.0)|
> |Sparse|x|x|x|**93.69**(+2.57)|**0.01**($\downarrow$ 28)|
> >
> >*SMR Component Analysis. Results are benchmarked on the PROXD dataset. POSA is used as a backbone. The kept 3-mask setup is used when Sparse Masks are available.*

---

> ### Author Response · Authors · 2023-11-21
> **Looking foward to your feedback**
>
> Dear Reviewer **gK1o**,
>
> Many thanks for your time and feedback on our paper. As the authors-reviewers discussion will end soon, please let us know if you have any further questions or comments.
>
> Best regards,

---

> > ### Comment · Reviewer_gK1o · 2023-11-22
> >
> > Thank you for your detailed responses. The added SMR Component Analysis is helpful in fully understanding the source of the improvements in accuracy and acceleration. However, I'm still not convinced that the contribution reaches the bar of acceptance with major concerns lying in novelty and contribution
> >
> > 1. First, the major novelty lies in the sparse mask selection for HSI, which is incremental upon the Choy 2020 citation, instead of sparse mask representation, which has been covered in Choy 2020, making the title a little over-selling.
> >
> > 2. Even though, it would have convinced me more if it could be formulated as a low-cost plug-in for existing methods. However, as discussed in Sec. 5, the compatibility and cost are limited.
> >
> > 3. As for contribution, the improvement in accuracy is promising and inspiring for HSI, while acceleration appears to be a less crucial characteristic for HSI as I understand. I would expect more positive characteristics that sparsity could bring specifically to HSI understanding. A missing opportunity would be the sparse mask distribution with respect to the human body structure. Similar knowledge would be more helpful to the community than the new method.
> >
> > Due to the above concerns, I tend to keep my rating.

---

> > > ### Author Response · Authors · 2023-11-23
> > > **Response for first comment from Reviewer gK1o.**
> > >
> > > Thank you for your reply. About the novelty of the paper, we believe that our method is not an extension of Choy et al. since we are focusing on sparse masks while only leveraging the COO format for transforming inputs. About compatibility, as mentioned in our Limitation (Section 5), our method can not adapt to models that add noises to deal with their tasks since noises may mislead the limited data in sparse inputs. Besides them, our proposed SMR can be applied easily in other methods. About the correlation between sparse distribution with respect to the human body structure, it is evident that vertices in many human joints are redundant and are not taken into account in our proposed SMR method, compared with the baseline POSA that still considers this information and gives incorrect predictions in contact points. This evidence can be viewed in Figure 3, Table 4, Figure 11, and Section E.4. We will have an additional heat map to clearly verify the sparsity distribution over the human body. Again, thank you for your response.

---

### Official Review · Reviewer_t62A · 2023-10-31

**Soundness:** 3 good
**Presentation:** 3 good
**Contribution:** 3 good
**Rating:** 6
**Confidence:** 4

**Summary:**

The paper introduces a novel approach called Sparse Mask Representation (SMR) to effectively handle the complex and sparse input data in human-scene interaction. Unlike previous methods that focused on lightweight models or quantization, SMR uses sparse masks to select important information from the input, reducing computational cost. Experimental results demonstrate its superior performance in contact prediction and scene synthesis tasks, with significantly faster inference speed.

**Strengths:**

The paper is skillfully written, ensuring ease of comprehension for the reader. It introduces a seemingly straightforward yet remarkably effective solution to the complex issue of human contact prediction in 3D environments. The authors have undertaken a thorough set of experiments, clearly demonstrating the superior performance of their approach. Furthermore, they have meticulously examined the related work, providing a comprehensive comparison with existing literature across multiple datasets.

**Weaknesses:**

I find the paper to be well-written, and the authors have conducted thorough experiments to demonstrate the effectiveness of their approach. The only potential area for improvement lies in providing more detailed explanations on how the sparse masks are defined, especially in the context of task dependency. This would further enhance the clarity and depth of the paper.

**Questions:**

Are the sparse masks randomly generated, meaning do the 0 and 1 values occur at random locations? Or are the masks specifically tailored to the task at hand?

How does your approach handle fine-grained contacts, such as situations where the tips of the fingers come into contact with other objects?

Additionally, could you elaborate on how your approach addresses videos?

---

> ### Author Response · Authors · 2023-11-19
> **Response for Reviewer t62A**
>
> **Q1: How the sparse masks are defined, especially in the context of task dependency?**
>
> >The sparse masks are randomly generated based on the input format used for human-scene contact prediction. Each mask is a tensor whose shape is similar to the shape of the input. In the contact prediction task, each input is represented by using a tensor with shape $N_v \times N_S$, where $N_v$ is the number of vertices to represent the human mesh, and $N_S$ is the number of features (coordinate and contact label) of each vertex.
>
> **Q2: Are the sparse masks randomly generated, meaning do the 0 and 1 values occur at random locations? Or are the masks specifically tailored to the task at hand?**
>
> >Yes, sparse masks are randomly generated. During the learning process, the model relies on the mask score $\alpha$ to identify the contribution of masks. Only masks that show a high enough contribution are kept.
>
> **Q3: How does your approach handle fine-grained contacts, such as situations where the tips of the fingers come into contact with other objects?**
>
> >The datasets used in our task (PROXD, GIMO, BEHAVE) do not consider fine-grained contacts. For instance, a part of the human body, such as a hand or foot, is assumed to make contact with at most one object. We believe our method will work effortlessly with fine-grained contacts, however, the current limitation in datasets does not allow us to verify the results of this case.
>
> **Q4: Could you elaborate on how your approach addresses videos?**
>
> >Our contribution is the sparse mask representation, and it works both with frames or video-based backbone. For example, to manage consecutive frames in the Scene Synthesis task, we simply employ the algorithm-based ContactFormer backbone. In a standard configuration, ContactFormer typically receives only one input from the contact predictor to compute contextual dependencies and the dynamic evolution of interactions. However, in our proposed SMR, ContactFormer can consider multiple contacts with associated weights. This capability enables it to establish more robust contextual dependencies, leading to improved reconstruction results. A recap of our approach and other methods in the mentioned task can be found in Section A.2 of our Appendix.

---

> > ### Author Response · Authors · 2023-11-21
> > **Please let us know if you have futher questions**
> >
> > Dear Reviewer **t62A**,
> >
> > Many thanks for your feedback and questions during the initial review. Please do let us know if you have any further questions.
> >
> > Best regards,
> > Authors

---

### Official Review · Reviewer_fpDr · 2023-11-01

**Soundness:** 2 fair
**Presentation:** 3 good
**Contribution:** 2 fair
**Rating:** 6
**Confidence:** 2

**Summary:**

This paper proposes a new representation for human-scene interaction task. The authors suggest to inject sparsity in the input space rather than designing lightweight model, model pruning or quantization which were used by previous methods. By enforcing input sparsity, the method is simple and effective in benefitting both the accuracy and inference time.

**Strengths:**

* Proposed approach is very simple but effective. The design choices made by the authors are intuitive and make sense.
* The experiments and analysis are quite comprehensive and provides insights for the method.
* Discussion is fairly done and includes a number of limitations and future directions. Overall it is a well written research paper.

**Weaknesses:**

* Methodology is incremental and not much novelty by itself.

**Questions:**

I can see that from Figure 6. it shows optimal performance for k = 3 mask with 90% sparsity, but there is no clear pattern or correlation between sparsity ratio vs accuracy. Could authors give an explanation on this trend?

---

> ### Author Response · Authors · 2023-11-19
> **Respone for Reviewer fpDr**
>
> **Q1: About the methodology contribution.**
>
> We thank the reviewer for the comment, we verify our contribution as below:
> > - We propose a simple yet effective method that focuses on ***the representation of inputs used for human-scene interaction***. Our method reduces processing time while enhancing model effectiveness by eliminating redundant information. Existing works on human-scene interaction have not yet investigated this problem.
> > - We not only utilize sparse masks as in previous work but also propose a refinement strategy to create zero-filtered inputs.
> > - Despite simplicity, our method outperforms other state-of-the-art methods significantly. For example, our method outperforms MIME [1] with ***2.72% higher accuracy and 60 times faster***.
>
>
> **Q2: I can see that from Figure 6. it shows optimal performance for k = 3 mask with 90% sparsity, but there is no clear pattern or correlation between sparsity ratio vs accuracy. Could authors give an explanation on this trend?**
>
> > We appreciate the reviewer's feedback. In the original submission, we have shown the correlation between the sparsity ratio, number of masks, accuracy, and running time in Figure 6. That figure shows that using only one mask results in a significant drop in performance due to missing information. Increasing the number of masks helps retain essential information and enhances the overall model performance. However, an excessive number of sparse masks (50 masks) can introduce redundant and inconsistent information, potentially lowering the overall accuracy. By maintaining the sparse masks at appropriate values (which can be easily achieved using Sparse Mask Refinement), we can optimize performance while significantly speeding up the evaluation process.
>
>
> >As per your suggestion, we have further added a comprehensive discussion about the mentioned information in Section E.1 of our Appendix, providing a more comprehensive exploration. This includes an in-depth analysis of the correlation between the number of sparse masks and sparsity concerning accuracy. Additionally, another experiment examining inference speed is also conducted within this section.

---

> ### Author Response · Authors · 2023-11-21
> **Please let us know if you have futher questions**
>
> Dear Reviewer **fpDr**,
>
> Please do let us know if you have any further questions before the authors-reviewers discussion period.
>
> Best regards,
> Authors

---

### Author Response · Authors · 2023-11-19
**Respone for All Reviewers**

## General Response
Dear ACs and Reviewers,

Thanks for your valuable reviews and insightful comments, which have helped us improve our paper. During the initial reviews, Reviewers **fpDr** and **t62A** were inclined toward acceptance. We are glad that our proposed sparse mask representation for human-scene interaction is a "simple/straightforward yet remarkably effective solution" (Reviewer **fpDr** and **t62A**), "is novel and technically sound" in Human-scene interaction task (Reviewer **BCwn**), and "benefits both the accuracy and inference time" (Reviewer **fpDr** and **gK1o**). We are also encouraged that our paper "is easy to read and skillfully written" (Reviewer **BCwn**, **fpDr**, and **t62A**), "provide a comprehensive comparison" (Reviewer **t62A** and **fpDr**), achieve "superior performance" (Reviewer **gK1o**, **t62A** ).

The common concern raised by Reviewers is the clarification of our contribution (Reviewer **fpDr**, **gK1o**, **BCwn**). We agree that Sparse Coding is not a new direction, as it has been mentioned in the Related Work section. However, the effectiveness of this approach ***remains unexplored*** in human-scene interaction tasks. Regarding the use COO format, we show that the COO is the right implementation choice for our sparse mask representation. Nevertheless, we believe that *using the right tool and contributing a simple, but effective method is more suitable than over-complicated things or reinventing the wheel.*

Despite the simplicity (and in many cases being considered as "limited novelty"), our method shows significantly strong results. For example, our method outperforms recent work MIME [1] with ***2.72% higher accuracy and 60 times faster*** (Table 1 in our main paper). Finally, since ICLR is a "Learning Representations" conference, we believe that our sparse mask representation technique is well-fit and would be valuable for the community despite its simplicity.



We are looking forward to responding to any further questions you have on our submission.

## Summary of Revision
Integrating the suggestions and feedback from all reviewers, besides fixing typos, figures, and adding recommended citations, we have made the following updates (yellow highlight) in the revision.
- We have further explained the correlation analysis between sparse masks and sparsity ratios in terms of accuracy and speed in Section E.1 of the Appendix (suggested by Reviewer **fpDr**).
- We have added SMR Component Analysis to clearly identify the contribution gained from different parts of the proposed method in Section E.2 of the Appendix (suggested by Reviewer **gK1o**).
- We have incorporated the result comparison between our proposed SMR and other mesh subsampling methods in Section E.3 of the Appendix (suggested by Reviewer **BCwn**).
- We have added an extended analysis of how sparse marks deal with redundant information in Section E.4 of the Appendix, which improves the model performance (suggested by Reviewer **BCwn**).


## Reference
[1] H. Yi, C.-H. P. Huang, S. Tripathi, L. Hering, J. Thies, and M. J. Black. Mime: Human-aware 3d scene generation. In CVPR, 2023.

---

> ### Author Response · Authors · 2023-11-20
> **A friendly reminder**
>
> Dear Reviewers **fpDr**, **t62A**, **gK1o**, **BCwn**
>
> We sincerely appreciate the time and effort throughout the reviewing process of our submission. As the author-reviewer discussion is due soon, please let us know if you have further questions about our submission.
>
> Once again, thank you in advance, and we look forward to your feedback.
>
> Best regards,
> Authors.

---

### Meta-Review · Area_Chair_uEbp · 2023-12-09

**Metareview:**

The paper presents Sparse Mask Representation, an approach that increases the inference speed and effectiveness of human-scene interaction models by converting high-dimensional inputs into sparse tensors, demonstrating significant improvements in accuracy and inference time over current methods in tasks like contact prediction and scene synthesis. Two reviewers recommend rejecting the paper and two are giving borderline acceptance. The main reason for both reviewers to reject the paper is the contributions over previous work, especially Choy 2020, are not very significant. Reviewer t62A who gave borderline acceptance also agrees with this limitation. After carefully reading the discussion between the reviewers and the authors, and reading the previous manuscript. The AC recommends rejecting this paper.

**Justification For Why Not Higher Score:**

See metareview above

**Justification For Why Not Lower Score:**

N/A

---

### Decision · Program_Chairs · 2024-01-16

Reject